# Effect of Brain Lesions on Voluntary Cough in Patients with Supratentorial Stroke: An Observational Study

**DOI:** 10.3390/brainsci10090627

**Published:** 2020-09-10

**Authors:** Kyoung Bo Lee, Seong Hoon Lim, Geun-Young Park, Sun Im

**Affiliations:** 1Department of Rehabilitation Medicine, St. Vincent’s Hospital, College of Medicine, The Catholic University of Korea, Seoul 06591, Korea; kblee0732@naver.com (K.B.L.); limseonghoon@gmail.com (S.H.L.); 2Department of Rehabilitation Medicine, Bucheon St. Mary’s Hospital, College of Medicine, The Catholic University of Korea, Seoul 06591, Korea; rootpmr@catholic.ac.kr

**Keywords:** Cough, mechanisms, pneumonia, stroke, brain mapping

## Abstract

Patients with stroke are known to manifest a decreased cough force, which is associated with an increased risk of aspiration. Specific brain lesions have been linked to impaired reflexive coughing. However, few studies have investigated whether specific stroke lesions are associated with impaired voluntary cough. Here, we studied the effects of stroke lesions on voluntary cough using voxel-based lesion-symptom mapping (VLSM). In this retrospective cross-sectional study, the peak cough flow was measured in patients who complained of weak cough (*n* = 39) after supratentorial lesions. Brain lesions were visualized via magnetic resonance imaging (MRI) at the onset of stroke. These lesions were studied using VLSM. The VLSM method with non-parametric mapping revealed that lesions in the sub-gyral frontal lobe and superior longitudinal and posterior corona radiata were associated with a weak cough flow. In addition, lesions in the inferior parietal and temporal lobes and both the superior and mid-temporal gyrus were associated with a weak peak cough flow during voluntary coughing. This study identified several brain lesions underlying impaired voluntary cough. The results might be useful in predicting those at risk of poor cough function and may improve the prognosis of patients at increased risk of respiratory complications after a stroke.

## 1. Introduction

Respiratory dysfunction and pneumonia are the leading causes of post-stroke hospitalization and high mortality [1]. Many patients with respiratory complications may manifest impaired cough reflex and a poor clearance of secretions, along with impairments in swallowing, which may increase the risk of aspiration. Although coughing and swallowing involve coordinated but distinct oral, laryngeal, and respiratory muscles [2], they both share similar nerves and muscles with a significant overlap in their control mechanisms. Despite these similarities, it is still difficult to predict patients at risk of impaired cough, especially in those with swallowing disturbances. The early identification of those with impaired cough and, thus, at risk of respiratory complications based on neuroimaging findings might be helpful.

The motor act of cough, which is characterized by the reorganization of the central breathing pattern to produce the three characteristic phases of a typical cough (inspiration, compression with glottic closure, and expiration) is modulated by the cerebral cortex [2]. A cough may be produced in a reflexive or voluntary manner, although both types prevent aspiration and protect the airway. Reflexive cough (RC), which is automatically generated by afferent activation, is involved in protecting the airway from aspiration during swallowing and is controlled by the brainstem. The voluntary cough (VC) is a conscious act that requires the voluntary activation of the respiratory muscles, including the diaphragm, which is a major inspiratory muscle, and has been shown to be modulated by the cerebral cortices. Stroke patients are known to have a reduced corticorespiratory drive, leading to weakness of the respiratory muscles, including the diaphragm, and which subsequently leads to a poor peak cough flow (PCF) during VC [3,4].

While the cough center is located in the brainstem, previous transcranial magnetic stimulation studies have shown that the excursion of the diaphragm, which is important in VC, is mediated by cortico-respiratory projections across the cortex [5,6,7], suggesting supra-medullary control. In addition, experiments have demonstrated that coughing is elicited after the electrical stimulation of the cortices, such as at the suprasylvian gyrus [8] or supplementary motor areas [9]. Although the middle cerebral artery (MCA) [10,11] and capsular infarctions [12] represent major risk factors contributing to a weak diaphragm, the localization of supratentorial regions associated with poor VC is still not well known [13]. The supramedullary control and brain lesions associated with poor VC are still unknown.

A decrease in VC indicates a decreased ability to clear the airways of aspirate material or mucus or secretion from the lower airway, which may predispose patients to aspiration pneumonia. Aspiration is common after stroke and is associated with an 11-fold increase in the risk of chest infections [14]. Therefore, the accurate identification of those at risk of poor cough based on brain regions may help to identify those at risk of aspiration pneumonia and who would benefit from an early implementation of adequate respiratory training. These trainings to accommodate the weak VC in stroke patients can lead to improved respiratory muscle strength and coughing skills, reducing the risk of aspiration [15].

Voxel-based lesion-symptom mapping is a neuroimaging technique that can be used to study the relationship between tissue damage and behavior on a voxel-by-voxel basis. VLSM has been widely used to identify lesion-symptom relationships in stroke patients, and has been used extensively to study, among other functions, language [16], gait [17], and spasticity [18]. A recent study a recent study by Suntrup-Krueger et al. [19] demonstrated via VLSM the role of specific cortical lesions with impaired RC in stroke patients. However, VC was not included in the VLSM analysis. It is well known that VC is cortically mediated, but which of the supratentorial lesions is associated with impaired VC is yet to be determined. In addition, whether or not these regions overlap with those reported for the RC needs to be elucidated.

Thus, the aims of this study were to investigate the effects of stroke lesions on VC function using VLSM and determine whether distinctive brain lesions are associated with poor VC in post-stroke patients. We hypothesized that VC would be associated with specific supramedullary brain regions that have been linked with cortico-respiratory projections. In light of the close relationship between VC and swallowing we also aimed to determine if those with poor VC manifest with poor swallowing function. Post-stroke patients with dysphagia are known to manifest with decreased airway clearance, and compromise of the respiratory muscles involved in VC. Therefore, we also hypothesized that those with reduced VC and decreased airway clearance would be associated with more pronounced difficulties in swallowing.

## 2. Materials and Methods

### 2.1. Patient Selection

This was a retrospective cross-sectional study that included 39 right-handed patients with confirmed ischemic stroke who showed a weak cough or a decreased ability to clear the throat. The medical records of patients with a record of weak VC, based on the assessment of a physiotherapist at the point of rehabilitation entry, were used for data collection [20]. The medical records of these participants were retrospectively reviewed.

Stroke lesions were confirmed on the basis of neurological symptoms and the results of the initial brain imaging studies, such as magnetic resonance imaging (MRI) or computerized tomography (CT) scan. All the data were obtained from patients recruited from a single stroke center from August 2015 to July 2017. However, because the main objective of this study was to evaluate the supramedullary brain control in VC, only those with ischemic stroke confined to the supratentorial region who met the following inclusion criteria were recruited into the study: (1) age from 20 to 80 years; (2) first-ever unilateral stroke; (3) ability to follow verbal instructions; and (4) full records of swallowing and respiratory pressure measurements, including voluntary PCF.

The exclusion criteria were as follows: (1) a history of spinal cord injury affecting the respiratory muscles; (2) stroke related to infratentorial lesions; (3) a history of chronic obstructive pulmonary disease, asthma, or other lung disorders that require oxygen therapy and affect respiratory pressure parameters; (4) Parkinson’s disease, myopathy, head and neck cancer lesions, dementia, or other disorders that affect respiratory function; and (5) poor conscious state or severe cognitive or language dysfunction that results in the inability to follow commands to initiate a cough.

To meet the objectives of this study, only a homogenous group of ischemic stroke patients were included, and those with intracerebral hemorrhage and bilateral brain lesions were excluded, as was carried out by previous VLSM studies [16,18,21]. The edema and brain shifting in the former group and the more severe neurological deficits in the latter group can mislead VLSM analysis.

Demographic and brain MRI data were collected from all the subjects to evaluate swallowing and respiratory pressure parameters. Brain lesions were evaluated using a high-resolution 3-T anatomical MRI system with a 5 mm slice thickness within 14 days of stroke.

The study protocol was reviewed and approved by the Institutional Review Board of our institution (Registry No. BC000000).

### 2.2. Clinical Assessment

The basic patient demographics and medical diagnosis, along with the National Institutes of Health Stroke Scale (NIHSS) score [22] and stroke classification according to the Trial of ORG 10172 in Acute Stroke Treatment (TOAST) criteria [23] were recorded. Data pertaining to functional assessments, including the modified Barthel Index (MBI) [24] and Mini Mental State Examination (MMSE) results, were retrieved from the medical charts.

### 2.3. Lesion Analysis

Lesions were mapped using the MRIcron (https://www.nitrc.org/projects/mricron) and were drawn manually on individual T2 scans of patients by a trained image analyst, and confirmed by an experienced clinical psychiatrist, who was blinded to all clinical data except for the affected side of the patients with hemiparesis. For more accurate analyses, the origin of each image (coordinates: 0 × 0 × 0 mm) was reoriented such that it was located close to the anterior commissure. To increase the statistical power for the identification of the lesion pattern, which showed a significant contribution to PCF independent of hemispheric lateralization, the volume-of-interest images were transformed to the right hemisphere. To analyze the mutual lesion maps, segmentation and normalization were employed [17]. We used the MR segment-normalized function of a plugin toolbox (https://www.nitrc.org/projects/clinicaltbx/) to map the stereotaxic space using the normalization algorithm provided by the SPM8 (http://www.fil.ion.ucl.ac.uk/spm/software/spm8) software. T2 images were co-registered with each participant’s T1 MRI. The T1 and lesion maps were normalized to the Montreal Neurologic Institute space using statistical parametric mapping. A unified segmentation-normalization was performed on the anatomical scan. A VLSM procedure was developed to determine the relationship between tissue damage and behavior on a voxel-by-voxel basis. VLSM was usually performed using binary data (with/without a deficit) with a cutoff value. However, information reflecting varying degrees of capacity may be lost in such an approach [16]. To avoid this potential issue, a direct statistical comparison of lesions was performed according to the degree of PCF using a VLSM method implemented in non-parametric mapping (NPM) software included in the MRIcron software [25]. Only voxels that exhibited lesions in at least 10% (*n* = 4) of all patients were included in the final analysis. The non-parametric Brunner–Munzel test for continuous data was used [25] because of the continuous clinical deficit. In the NPM analyses, a lower value refers to a poorer performance; thus, subjects with a score of 40 were more severely affected compared with those scoring 80. Colored VLSM maps representing the z statistics were generated and overlaid onto the automated anatomical labeling and Johns Hopkins University white matter templates provided with the MRIcron software [25]. Additionally, to identify relevant anatomic structures implicated in the analyses, Talairach Daemon (http://www.talairach.org/client.html) was used [26]. *p*-values < 0.05 were considered to indicate statistical significance.

### 2.4. Respiratory Pressure Parameters

To assess the voluntary PCF, the patient was asked to perform a quick forceful cough. The clinician educated the patient regarding how to cough on the portable spirometer. Both voluntary and respiratory pressure parameters were measured by therapists not involved in the imaging analysis of the participants.

The voluntary PCF, the maximal inspiratory pressure (MIP), and the maximal expiratory pressure (MEP) were measured using the same methods following the guidelines recommended by the American Thoracic Society/European Respiratory Society [27]. For VC, the PCF value for each patient was presented as the mean of the three highest values over five consecutive attempts.

The MIP and the MEP were measured using a respiratory pressure meter (Micro-Plus Spirometer; Carefusion, Corp., San Diego, CA, USA) with a standard flange mouthpiece. For those with facial palsy and were unable to perform a perfect lip seal, the therapist aided by holding the lips around the mouthpiece between the lips and nozzle to minimize air leak. The highest recorded values after three attempts were used for the analysis. PCF was used to measure the VC (L/min). All the subjects were asked to perform a quick, short, and explosive cough on a peak flow meter (Micropeak; Carefusion, Corp., San Diego, CA, USA) to a face mask. Before the patients’ voluntary PCFs were formally measured, verbal instructions were given by the clinicians on the procedures of how to cough in response to a command. The clinician then performed a demonstration of how to cough on the portable spirometer in front of the patient. In those with poor comprehension capabilities, the patient was allowed to practice with the clinician a few times before the formal assessment [28].

Among the participants, those with an increased risk of aspiration were defined as patients with PCF values lower than <80 L/min obtained from the spirometry assessment. The cutoff values were based on a previous report that a VC below this value increases the risk of respiratory complications by 9.2 (95% CI: 3.5–23.9) and can predict respiratory complications through a multivariate analysis with an area under the curve of 0.8 (0.8–0.91) [28].

Aspiration pneumonia within the first month after stroke onset was identified via a retrospective review of the medical records. Aspiration pneumonia was defined by respiratory symptoms with a temperature exceeding 38 °C, leukocytosis, and infiltration confirmed by chest radiography, warranting the use of antibiotics [29].

### 2.5. Swallowing Assessment

All the participants underwent a formal swallowing assessment first, via a screening test—the Gugging Swallowing Screen (GUSS) test [30,31] at admission, followed by formal assessment using the Mann Assessment of Swallowing Ability (MASA) scales [32], the Functional Oral Intake Scale (FOIS) [33], and the Penetration-Aspiration Scale (PAS) [34] scores obtained during the initial videofluoroscopy studies (VFSS). The FOIS was obtained at the time of the VFSS, which was performed according to standard protocols [35]. For the PAS, the worst PAS score across all boluses were used for analysis [36]. All the assessments were performed within the first two weeks of acute stroke.

The diagnostic properties of these parameters have been demonstrated in previous studies, which have shown excellent levels of the inter-rater reliability of the MASA (Kappa = 0.82) [37], GUSS (agreement level = 83%) [30,31], and FOIS (agreement level = 85%) [34] scales, respectively.

### 2.6. Statistical Analysis

Intergroup differences between those with PCF <80 and ≥80 L/min were assessed using the *t*-test and chi-squared test, as appropriate, based on the data expressed as either the mean or median values for continuous variables and frequencies and percentages for normal variables. If the Kolmogorov–Smirnov test and the Shapiro–Wilk test failed to show a normal distribution, the Mann–Whitney U test was used in place.

A correlation analysis of the variables between the PCF and swallowing parameters was performed, followed by an analysis of the volume lesion and functional parameters. The rho values were considered to show the following levels of correlation: ranging between 0.9 and 1.0 for a very high positive correlation; 0.7–0.8, high positive correlation; 0.3–0.6, low positive; and 0.0–0.2, with negligible correlation. All *p*-values < 0.05 were considered significant. Statistical analyses were performed using the R 2.15.3 package software (R Foundation for Statistical Computing, Vienna, Austria).

## 3. Results

### 3.1. Clinical Assessment

A total of 39 patients that fulfilled the inclusion criteria and underwent full assessment using the respiratory pressure meter were identified. The basic demographic and clinical features of the patients are presented in Table 1.

### 3.2. Lesion Analysis

A lesion overlay of all the subjects is presented in Figure 1. Based on the results of the VLSM with NPM, lesions of the subgyral parietal and frontal lobe, superior longitudinal fasciculus (SLF), posterior corona radiata, temporal lobe, the posterior limb of the internal capsule, and the superior temporal gyrus (STG) of the temporal lobes were associated with decreased post-stroke coughing function (Figure 2, Table 2).

### 3.3. Respiratory Pressure Parameters

All the assessments were made at 9.4 ± 6.8 days after stroke onset. The mean value of PCF was 98.4 ± 58.4 L/min. Those with low cough values (*n* = 19) with increased risk of aspiration showed significant differences in the stroke severity and functional disabilities.

Twelve of the 39 patients (30.7%) had a positive history of aspiration pneumonia during the first month of stroke onset. Those with a PCF < 80 L/min (52.6%) showed a higher proportion of patients with aspiration pneumonia than those with ≥ 80 L/min (10%) within the first month after onset (*p* = 0.004).

### 3.4. Swallowing Parameters

Overall, most patients had some evidence of dysphagia, but those with a PCF < 80 L/min showed a higher severity of dysphagia. By the time of the assessments, their median FOIS value was “1,” which denoted the nil per os status. However, no intergroup differences were observed in the PAS scores.

### 3.5. Correlation Analysis of PCF

The PCF showed a statistically significant (*p* < 0.001) positive correlation with the swallowing parameters that included the FOIS (*r* = 0.695), MASA (*r* = 0.714), and GUSS (*r* = 0.733) scores, confirming that a weak cough was correlated with an increased severity of swallowing disturbance (Figure 3). However, no significant correlation was observed with the PAS (*r* = −0.221). Modest degree correlations were also found between the PCF and other functional parameters (i.e., NIHSS, Berg, MMSE, MBI) (absolute rho values = 0.38–0.58).

### 3.6. Correlation Analysis of Lesion Volume

There were no significant differences between the lesion volume of patients with a “weak” versus “strong” cough (*p* = 0.766). A correlation analysis of the lesion volumes and PCF failed to show any significance (rho = −0.25, *p* = 0.13). Although the lesion volume showed some significant correlations with the MEP (*p* = 0.016) and MIP (*p* = 0.022), the rho values indicated only a moderate negative association (rho= −0.38, *p* = −0.36). In addition, the lesion volume showed significant correlations with other functional parameters such as the MBI (rho = −0.357, *p* = 0.025) and initial NIHSS scores (rho = 0.53936, *p* = 0.00039) (Figure 4).

## 4. Discussion

The results of this VLSM study suggest that specific supratentorial lesions may be linked with decreased subjective cough weakness in patients diagnosed with stroke, confirming the findings of previous studies [6,7,11] implicating both cortical and subcortical regions in the control of VC. Our findings establish the role of supratentorial lesions in the frontal sub-gyral area, the STG, some parts of the parietal lobes, and the superior corona radiata. In addition, the SLF showed positive association. Knowledge of these brain sites involved in poor VC may facilitate the early identification of patients with poor coughing function. Among those with subjective cough weakness, those at risk of aspiration showed an increased severity of swallowing, confirming the close link between swallowing and cough.

Owing to the disruption of the cortical and medullary areas associated with cough generation [38], both the RC and VC may be affected after stroke [10]. These two types of cough share common features, including a 3-phase inspiration pattern compressed with glottic adduction followed by expulsion via the contraction of the respiratory muscles [39]. They also share the same efferent. In line with these similarities, a few lesions from our study overlapped with those known to be involved in poor RC [19]. The overlapping lesions included the temporal lobe and STG areas, which are also crucial components of the swallowing mechanism [19]. The supramarginal gyrus and temporal area represent sensorimotor regions and are known to be associated with impaired swallowing and also with impaired motor reaction in RC [19]. The similarities of the muscle activation and output between RC and VC suggest a possible role for the temporal, STG, and supramarginal gyrus lesions in the motor components of cough [19], and their potential crucial roles in airway protection during swallowing.

Despite these similarities, the underlying musculoskeletal mechanisms and motor patterns of these two coughs differ considerably [39], with different stroke sites resulting in distinct cough impairments. Specific brain lesions that were exclusively investigated in this study include the frontal lobe, the posterior corona radiata, and the sub-gyral area, which consist of descending cortico-respiratory projections located within the pyramidal tract [12]. These cortico-respiratory projections are frequently affected in patients with hemiparesis due to stroke [6,7]. These regions have not been strongly implicated in RC impairment in previous studies [19]. Our findings support the clinical role of these regions in the diminished ability to generate a VC due to direct involvement of the cortico-respiratory tract.

The frontal subgyral region and the frontal cortex may correspond closely to the cortical “hotspot” sites identified by electrophysiological studies using transcranial magnetic stimulation [5,6,12]. Furthermore, the cortical representation of the inspiratory muscles is known to lie close to the vertex [40]. The corona radiata is part of the descending cortico-respiratory projection located within the pyramidal tract [12], and is also associated with an increased risk of aspiration [21]. Although swallowing has been related to the superior or anterior portion of the corona radiata, our results indicate the involvement of the posterior part. The corona radiata is somatotopically arranged, and while the anterior part involves the corticobulbar tract, the posterior portion controls the truncal muscles [21]. Since the role of the truncal and abdominal muscles is related not only to truncal control, but also to the control of the respiratory pump muscles and ventilation [41], it is plausible that the posterior portion of the corona radiata may reflect the involvement of respiratory muscles in VC. Thus, although not directly part of the corticobulbar tract involved in the act of swallowing, our results further suggest that these posterior corona radiata regions are crucial regions in airway protection.

An unexpected finding was the SLF, which is not part of the cortico-respiratory projections, showing a positive association with VC. The SLF, which consists of longitudinal fibers that connect the dorsolateral frontal and parietal cortices [42], mediates the spatial coordination of the trunk and limbs and contributes to the preparatory stages of movement planning [43]. Therefore, the SLF may contribute to the preparatory truncal function related to the respiratory muscles, such as the abdominal and respiratory muscles required for coughing. The SLF also plays an important role in swallowing, and its temporal part has been implicated in impaired RC [19]. In addition to these past findings, our results demonstrate that lesions in the SLF may also interfere with airway protection by limiting VC.

Although swallowing and coughing are independent behaviors, both these actions are strongly coordinated structural movements that require the reconfiguration of the ventilatory breathing pattern [2]. Apparently, a few brain lesions already known to be involved in swallowing, such as the STG [19] and SLF [21], overlapped with lesions that were associated with a weak cough in this study. Therefore, it is not surprising that the PCF values showed strong correlations with those with weak coughs showing more severe swallowing impairments as assessed by several parameters, suggesting that swallowing and coughing are closely linked. One may cautiously suggest that the simultaneous involvement of some brain regions, such as the STG and SLF, could lead to both disordered swallowing and the poor production of a strong VC to clear the airways. The coordination of these two behaviors is vital to protect the airway from aspiration. Therefore, further studies are needed to understand coughing and swallowing as both independent and as coordinated responses to aspiration events [44].

Post-stroke patients are known to carry impaired contralateral corticodiaphragmatic pathways and might cause abnormal motion of the diaphragm. Patients diagnosed with dysphagia after stroke are known to exhibit a greater degree of diaphragm weakness than those without swallowing disturbances [3]. Accordingly, patients with stroke-related dysphagia are known to produce low PCF during VC compared to stroke patients without dysphagia and healthy controls [45]. Hence, our results showed that those with a PCF below 80 L/min showed an increased severity of dysphagia and more respiratory infections. These results are similar to those of studies that have shown that a low cough strength based on the measurement of cough flow can be used as an indicator of pneumonia risk and in acute stroke [46]. Respiration, swallowing, and coughing function all share similar neural and anatomical substrates. The close relationship between swallowing and protective cough is crucial, but the mechanism of these interconnected systems needs to be elucidated.

Increased lesion volume has been associated with severe dysphagia in previous studies [21]. In contrast, our results failed to show a significant association between brain volume and PCF, obviating the need for adjustment of stroke lesion volume in our analysis. Although the functional parameters were affected by the total stroke volume, PCF was more specific to the lesion location rather than the total volume per se.

This study’s limitations are as follows. First, it was a retrospective study, and many subjects who were too ill to undergo spirometry may not have been included. Second, the PCF and PAS scores showed a poor correlation. This may have been attributable to the fact that only patients with subjective weakness of cough were included in the analysis. In addition, since the worst PAS score was used for analysis, the results should be interpreted with caution. Also, the use of the correlation analysis, which cannot control for the covariables, may have been insufficient to elucidate the complex relationship between the PAS and PCF. In order to further delineate the exact correlation between PCF and PAS, future VLSM studies that include those with no cough disturbance are warranted. Subjective judgment of cough strength can be misleading [47,48], and prospective studies with large sample sizes that include those with no cough weakness are warranted. Nevertheless, patients with a lower PCF showed an increased incidence of respiratory complications, which was consistent with the results of multiple regression analysis reported by Sohn et al. [28]. Third, no laterality was considered in this study with all the lesions flipped to one side. Previous studies have shown that the majority of patients with left MCA [20] show a weak or absent VC and suggested a laterality in VC, with the left cerebral hemisphere playing a dominant role in the voluntary control of cough. However, other studies failed to associate RC with a specific hemisphere [19,49]. Indeed, patients with right supratentorial stroke were also identified in our study. The discrepancies between our findings and those that suggested left laterality could be attributed to the varying number of cases and different assessment methods for defining weak cough, with the subjective classification of cough as either weak or absent, was used in the previous report [20]. All our cases were first screened by a physiotherapist and declared to have a weak cough based on the same subjective classification as above, and then had undergone full objective measurements of the PCF. In addition, one has to consider that the diaphragm and voluntary respiratory movement are under the control of both hemispheres of the brain with no dominance [50]. Fourth, the mean PCF in our study was lower than those reported in post-stroke patients with no coughing dysfunction (mean 195 ± 67.1 L/min) or from healthy participants (253.2 ± 90.8 L/min) [3], confirming that all those in our analysis had impairment in VC. The lower values than those encountered in a previous report [46] may also be related to the higher age of patients in our group. However, because the PCFs from VC are measured by volitional tests, with the patient making a maximal effort during the spirometry [51], the older age distribution might have limited this effort in our patients. Aging has also been related to the increased sarcopenia of the diaphragm [52], which in turn could lead to poor respiratory motor recruitment, causing insufficient coughs that would allow not full clearance. Fifth, the FOIS was used as one of the swallowing parameters. Although the FOIS [33] was assessed at the time as VFSS, caution is needed in judging patients’ swallowing abilities based solely on this parameter, especially in acute stroke patients. To overcome this factor, the GUSS [30,31] and MASA scales [32,33,37], which show good diagnostic properties in acute stroke, were included in the study. Finally, it should be determined whether the proposed lesions were specific to coughing alone, given the overlap of patients with both dysphagia and cough dysfunction. However, our results did not reveal obvious lesions that are hallmarks of dysphagia, such as the insula or thalamus. Instead the lesions in our study closely corresponded to the cortico-respiratory pathways [5,12,53].

In conclusion, this study provides novel insights into the cortical and subcortical dimensions of VC. The lesions defined in this study might facilitate the stratification of patients at risk of impaired VC and respiratory complications, and thus identify candidates for respiratory training [54], along with conventional rehabilitation therapy after stroke.

## 5. Conclusions

Cough and the respiratory muscles involved in it might be under different control than the limbs. In post-stroke patients, specific regions were associated with impaired VC. The information provided in this study could allow the early identification of patients with deficits in generating a proper VC, and thus with limited airway protective function.

## Figures and Tables

**Figure 1 brainsci-10-00627-f001:**
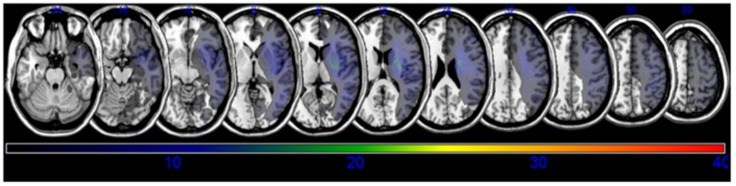
Overlay of lesions for stroke patients included in this study (*n* = 39). Maps are overlaid on a T1-template in Montreal Neurologic Institute space 1 × 1 × 1 mm.

**Figure 2 brainsci-10-00627-f002:**
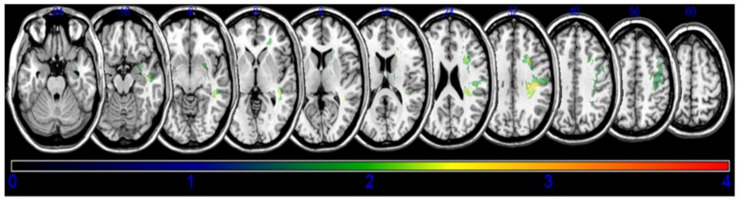
Statistical voxel-based lesion-symptom mapping. The nonparametric Brunner–Munzel statistical analysis was used for the continuous peak cough flow. Color scale indicates Brunner–Munzel rank order z-statistics. Only voxels significant at *p* < 0.05 are shown. Colored bar represents the z statistics. We set the maximum range of the Z score as 4, which are shown with the maximum brightness.

**Figure 3 brainsci-10-00627-f003:**
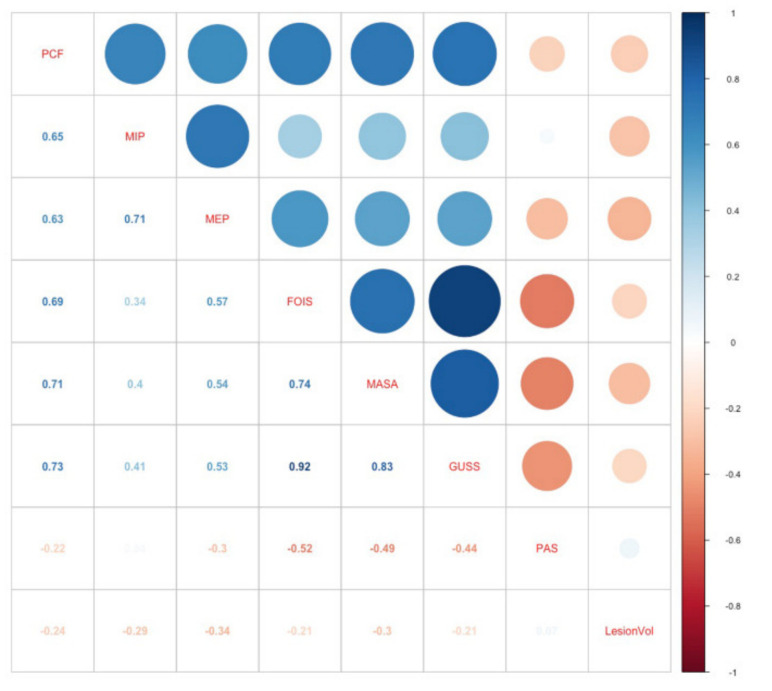
Heat map of the correlation coefficients (Spearman) between the peak cough flow (L/min) and the degree of dysphagia and aspiration, which are displayed in different colors. The color scale indicates the degree of correlation (blue, strong positive correlation; white, weak correlation; red, strong negative correlation). PCF: peak cough flow; MIP: maximal inspiratory pressure; MEP: maximal expiratory pressure; FOIS: Functional Oral Intake Scale; MASA: Mann Assessment of Swallowing Ability; GUSS: Gugging Swallowing Screen; PAS: Penetration Aspiration Scale; LesionVol: Lesion Volume.

**Figure 4 brainsci-10-00627-f004:**
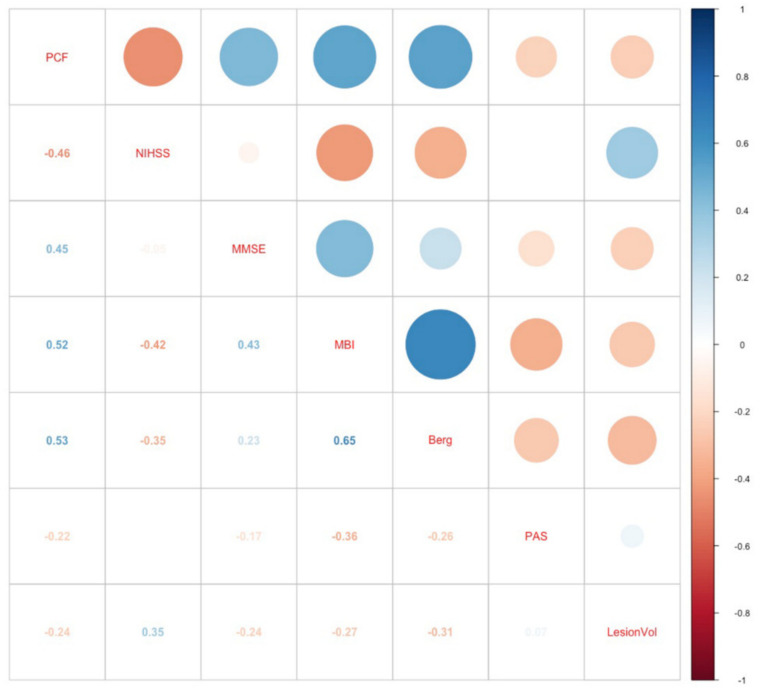
Heat map of correlation coefficients (Spearman) between the lesion volume and functional parameters displayed in different colors. The color scale indicates the degree of correlation (blue, strong positive correlation; white, weak correlation; red, strong negative correlation). PCF: peak cough flow; NIHSS: National Institutes of Health Stroke Scale; MMSE: Mini-Mental State Examination; MBI: modified Barthel Index; PAS: Penetration Aspiration Scale; LesionVol: Lesion Volume.

**Table 1 brainsci-10-00627-t001:** Basic demographic and clinical features.

Variables	Total (*n* = 39)	Cough <80 L/min (*n* = 19)	Cough ≥80 L/min (*n* = 20)	*p*-Value
Basic demographics				
Age	72.6 ± 12.5	76.6 ± 10.3	68.8 ± 13.5	0.0051
Gender				0.584
Male	25 (64.1)	13 (68.4)	12 (60.0)	
Female	14 (35.9)	6 (31.6)	8 (40.0)	
Body mass index (kg/m^2^)	22.9 ± 2.9	23.4 ± 2.6	22.4 ± 3.1	0.300
Brain lesion classification				
Total lesion volume (voxels)	55,985.1 ± 91,883.6	60,558.8 ± 92,950.9	51,640 ± 93,055.1	0.989
Laterality				0.648
Right	24 (61.5)	11 (57.9)	13 (65.0)	
Left	15 (38.5)	8 (42.1)	7 (35.0)	
TOAST				0.465
Large artery atherosclerosis	18 (46.1)	8 (42.1)	10 (50.0)	
Cardio embolism	9 (23.1)	6 (31.6)	3 (15.0)	
Small-vessel occlusion	12 (30.8)	5 (26.3)	7 (35.0)	
Medical comorbidities				
Diabetes mellitus	12 (30.8)	7 (36.8)	5 (25.0)	0.650
Hypertension	24 (61.5)	15 (78.9)	9 (45.0)	0.064
Atrial fibrillation	9 (23.1)	6 (31.6)	3 (15.0)	0.396
Hyperlipidemia	1 (2.6)	0 (0.0)	1 (5.0)	1.000
Neurological function				
NIHSS	6.4 ± 4.2	7.6 ± 4.2	5.3 ± 3.9	0.095
MBI	38.7 ± 27.7	26.5 ± 25.5	50.3 ± 25.0 ^1^	0.006
MMSE	19.6 ± 6.4	17.3 ± 6.6	21.8 ± 5.4 ^1^	0.0024
Berg	17.7 ± 20.6	9.7 ± 14.7	25.2 ± 22.8 ^1^	0.017
Respiratory pressure measurements				
Peak cough flow (L/min)	98.4 ± 58.4	53.0 ± 20.6	141.5 ± 48.9 ^1^	<0.001
MIP (cmH_2_O)	26.6 ± 25.4	14.5 ± 8.8	38.2 ± 30.5 ^1^	0.003
MEP (cmH_2_O)	40.2 ± 32.4	23.2 ± 16.6	56.4 ± 35.7 ^1^	0.001
Swallowing parameters				
FOIS	2 (1–4)	1 (1–2)	4 (2–4) ^1^	0.007
MASA	155.1 ± 17.1	142.9 ± 14.9	166.7 ± 9.3 ^1^	<0.001
GUSS	7.6 ± 4.3	4.7 ± 2.6	10.3 ± 3.7 ^1^	<0.001
PAS	8 (7–8)	8 (7.5–8)	7 (5–8)	0.137

Values are presented as mean ± standard deviation, number (%), and median (interquartile range). TOAST: Trial of ORG 10172 in Acute Stroke Treatment (TOAST) Classification; NIHSS: National Institutes of Health Stroke Scale; MBI: modified Barthel Index; MMSE: Mini-Mental State Examination; MIP: maximal inspiratory pressure; MEP: maximal expiratory pressure; FOIS: Functional Oral Intake Scale; MASA: Mann Assessment of Swallowing Ability; GUSS: Gugging Swallowing screen; PAS: Penetration Aspiration Scale. ^1^ Estimated by a *t*-test for continuous variables or Mann–Whitney U test between dough force < 80 L/min versus ≥ 80 L/min groups.

**Table 2 brainsci-10-00627-t002:** Stroke lesions related to voluntary cough impairment.

MNI Coordinates (X, Y, Z)	BM Z max	*n* Voxels	Anatomical Brain Lesion
34, −38, 35	2.90267	114	Parietal lobe, Sub-Gyral
29, −26, 32	3.38958	111	Frontal lobe, Superior longitudinal
29, 13, 28	2.85527	116	Frontal lobe, Sub-Gyral
29, −27, 30	3.17468	114	Posterior corona radiata
38, −35, 15	2.65207	106	Temporal lobe, STG
42, −41, −7	2.55235	115	Temporal lobe, MTG

The Montreal Neurological Institute (MNI) coordinates represents the voxels which is tested significant based on Brunner–Munzel (BM) Z score and the number (*n*) of clustering voxels that survived the threshold of *p* < 0.05, false discovery rate corrected. The anatomical region is identified using the Talairach daemon tool, the automated anatomical labeling and the Johns Hopkins University white matter templates. STG: superior temporal gyrus, MTG: middle temporal gyrus.

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
