# Peer review of "Effect of Brain Lesions on Voluntary Cough in Patients with Supratentorial Stroke: An Observational Study"

_brainsci, 2020, doi:10.3390/brainsci10090627_

Round 1

Reviewer 1 Report

Thank you for the opportunity to review this manuscript. The authors set out to “analyse the effects of stroke lesion on voluntary cough using voxel-based lesion-symptom mapping (VLSM)” in 39 patients with supratentorial lesions and subjectively weak cough. Using this method, the authors identified several brain areas associated with weak voluntary cough peak flow. The premise of this study is sound, as results have the potential to facilitate the earlier identification of patients at risk of reduced airway protection, and to inform future research efforts in this area. Unfortunately, recruitment bias (only patients with subjectively weak cough were recruited) represents a major limitation of this work. Regardless, there are several ways that I believe this manuscript could be strengthened, asoutlined below:

Introduction

  • P1 line 38: “The motor act of cough is characterized by reorganization of the central breathing pattern…which are known to originate in the cortex”. This statement is confusing, as the respiratory pattern for cough originates in the brainstem, not the cortex. The cortex certainly can modulate the brainstem-generated pattern (as when one is asked to “cough hard”) and is responsible for the initiation of voluntary coughing – is this what the authors meant? Please revise for accuracy, and consider revisiting the model of cough adapted by Troche et al.
  • P2 line 54: typo “accuracy identification”
  • P2 line 57: “identifying those at risk of poor cough is crucial to prevent aspiration pneumonia.” This argument is not well-explained…consider providing more details about the pathogenesis of aspiration pneumonia, and the relationship between cough and AP.
  • What were your hypotheses?

Methods

  • P2 line 72: typo “supratentorial are”
  • Inclusion criteria: please include the rationale for only including a) supratentorial stroke, b) ischemic stroke, c) unilateral stroke.
  • Was smoking an exclusion criteria? Were any aphasic patients excluded on the basis of instruction following?
  • Please provide more details about the portable spirometer – what make and model? Was a facemask used, or did performance require an adequate lip seal? Were there any participants who could not achieve an adequate lip seal?
  • Please provide a reference for your definition of aspiration pneumonia. Were all patients able to be followed up at 1 month for these symptoms, and how was this achieved?
  • Which patients underwent a swallowing evaluation – only the ones with a peak cough flow <80L/min? Analysis of swallowing was not stated as an aim of this study, so it is unclear why these measures were taken. Given that statistical comparisons were drawn between cough and swallowing outcomes, swallowing needs to be included in the primary aims. Please provide more details about the VFSS protocol e.g. number of boluses, consistencies, etc. This may help to explain why you did not find an association between PCF and PAS.

Results

  • Table 1 – please indicate whether the cough <80 L/min and the cough >80 L/min groups were significantly different on the variables presented in the table.
  • P6 line 200: Among patients who developed AP within the first month, 52.6% had a PCF < 80L/min. Does that mean that the other 57.4% had PCF > 80 L/min? I’m not sure what the “versus 10%” is referring to.
  • Should PAS also be included in figure 3?

Discussion

  • P8 line 239: I don’t agree with the authors’ conclusion that “results suggest that specific supratentorial lesions are associated with decreased VC function in patients diagnosed with stroke”, simply because only patients with subjectively weak VC were included in the analysis. It remains unknown whether, if patients with subjectively strong VC were also included, the same results would have been found. Please temper your conclusions accordingly.
  • P8 line 249: “They also share the same afferents and efferents” – this is not correct, as one can produce a VC in the absence of sensory stimulation.
  • P8 line 251: “The overlapping lesions include the temporal lobe and STG areas, which are also crucial components in the swallowing mechanism” – please provide a reference
  • A major limitation of this work is that only patients who were subjectively identified as having weak VC were included in the study, thus creating a potential exclusion bias. Clinical judgement of cough strength is not strong (Laciuga et al., 2016; Miles & Hucakbee, 20130. Please note this as a limitation in this section.
  • Another limitation is the use of FOIS as a measure of dysphagia severity. At the time of swallowing assessment, the median FOIS score equated to nil by mouth status – this is more reflective of subjective clinical judgement than actual swallowing ability, particularly as these patients were measured prior to VFSS. Please consider removing the FOIS as an outcome, or acknowledge that validity of the scale is grossly limited in acute stroke.

Author Response

September 3rd, 2020
The following are the responses to the comments raised by Reviewer 1.
Dear reviewer
We appreciate the comments raised during the review process. We have prepared a point-by-point response to the comments raised. The revised parts are marked with yellow highlight in the manuscript.
Comment
1.>P1 line 38: “The motor act of cough is characterized by the reorganization of the central breathing pattern…which are known to originate in the cortex”. This statement is confusing, as the respiratory pattern for cough originates in the brainstem, not the cortex. The cortex certainly can modulate the brainstem-generated pattern (as when one is asked to “cough hard”) and is responsible for the initiation of voluntary coughing – is this what the authors meant? Please revise for accuracy, and consider revisiting the model of cough adapted by Troche et al.
Authors’ response: We agree with the comments raised by the reviewer. We have revised the model adapted by Troche et al. and changed “…the motor act of cough is characterized by the reorganization of the central breathing pattern…which are known to be modulated by the cortex”.
Edited page: Page 1; line 40-42

2.> P2 line 54: typo “accuracy identification”
Authors’ response: The typo was corrected to “accurate identification”
Edited page: Page 2; line 62

3.> P2 line 57: “identifying those at risk of poor cough is crucial to prevent aspiration pneumonia.” This argument is not well-explained…consider providing more details about the pathogenesis of aspiration pneumonia, and the relationship between cough and AP.

Authors’ response: We agree that the relationship between the cough and aspiration pneumonia needs more elaboration. We have added an extra paragraph to describe in detail the pathogenesis of aspiration pneumonia and cough.
“A decrease in VC indicates a decreased ability to clear the airways of aspirate material, mucus or secretion from the lower airway which may predispose patients to aspiration pneumonia. Aspiration is common after stroke and is associated with an 11-fold increase in the risk of chest infections [14]. Therefore, accurate identification of those at risk of poor cough based on brain regions may help to identify those at risk of aspiration pneumonia and who would benefit from an early implementation of adequate respiratory training. These trainings to accommodate the weak VC in stroke patients can lead to improved respiratory muscle strength and coughing skills, reducing the risk of aspiration [15].”

Edited page: Page 2; line 59-66

4.>What were your hypotheses?
Authors’ response: We have described our hypotheses as follows
“Therefore, we hypothesized that VC would be associated with specific supramedullary brain regions that have been linked with cortico-respiratory projections. In light of the close relationship between VC and swallowing, we also sought to determine whether low VC can lead to more pronounced difficulties in swallowing parameters. Thus, the aims of this study were to investigate the effects of stroke lesions on VC function using VLSM, determine whether distinctive brain lesions are associated with poor VC in post-stroke patients, and describe the swallowing levels in those with low PCF during VC.”

Edited page: Page 2; line 76-81

Methods
5.>P2 line 72: typo “supratentorial are”

Authors’ response: The typo was corrected to “supratentorial region”
Edited page: Page 2; line 93

6. > Inclusion criteria: please include the rationale for only including a) supratentorial stroke, b) ischemic stroke, c) unilateral stroke.
Authors’ response: The main objective of this study was to determine the supramedullary control of cough in stroke patients; hence the rationale for including only those with supratentorial stroke. Homogenous ischemic stroke lesions are often included to delineate lesion-symptom relationship in VLSM analysis. Intracerebral haemorrhage can accompany with edema and brain shifting. Bilateral brain lesions result in more severe neurological deficits. These two heterogenic conditions can mislead VLSM analysis and hence the reason why only ischemic stroke patients were included in this study. The inclusion criteria employed in this study were also met by previous VLSM analysis in stroke patients.
Edited page: Page 3; line 103-106

7. > Was smoking an exclusion criteria? Were any aphasic patients excluded on the basis of instruction following?
Authors’ response: Smoking history by itself was not an exclusion criterion. Only those pulmonology patients that would “require oxygen therapy and affect respiratory pressure parameters” were excluded. As pointed by the reviewer, “poor conscious state or severe cognitive or language dysfunction that resulted in the inability to follow commands to initiate a cough” were also excluded. We have made these changes in the manuscript.

Edited page: Page 3; line 100-101

8.> Please provide more details about the portable spirometer – what make and model? Was a facemask used, or did performance require an adequate lip seal? Were there any participants who could not achieve an adequate lip seal?
Authors’ response: We have provided more information on the model and manufacturer of the portable spirometer. The PCF during VC was measured with a face mask. The MIP and MEP were measured using a standard flange mouthpiece. The details were provided as follows:
“The MIP and MEP were measured using a respiratory pressure meter (Micro-Plus Spirometer; Carefusion, Corp., San Diego, CA, USA) with a standard flange mouthpiece. The highest recorded values after three attempts were used for the analysis. The PCF was used to measure VC (L/min). All subjects were asked to perform a quick, short, and explosive cough on a peak flow meter (Micropeak) to a facemask. Before the patients’ voluntary PCFs were formally measured, verbal instructions were given by the clinicians on the procedures of how to cough in response to a command. The clinician then performed a demonstration of how to cough on the portable spirometer in front of the patient. In those with poor comprehension capabilities, the patient was allowed to practice with the clinician a few times before the formal assessment [29].”

Edited page: Page 3; line 159-168

9.>Please provide a reference for your definition of aspiration pneumonia. Were all patients able to be followed up at 1 month for these symptoms, and how was this achieved?
Authors’ response: The reference of aspiration pneumonia was from “Minnerup, J.; Wersching, H.; Brokinkel, B.; Dziewas, R.; Heuschmann, P.U.; Nabavi, D.G.; Ringelstein, E.B.; Schabitz, W.R.; Ritter, M.A. The impact of lesion location and lesion size on poststroke infection frequency. J. Neurol. Neurosurg. Psychiatry 2010, 81, 198-202” and was included in the manuscript. Aspiration pneumonia was identified via a retrospective review of the medical records. Our institution is a dedicated regional stroke centre, therefore patients even after early discharge to home or rehabilitation facilities, are able to make regular follow-ups or visit the department of infection should any fever or infection sign arise, therefore full medical records were retrievable.

Edited page: Page 5; line 174-177

10.>Which patients underwent a swallowing evaluation – only the ones with a peak cough flow <80L/min? Analysis of swallowing was not stated as an aim of this study, so it is unclear why these measures were taken. Given that statistical comparisons were drawn between a cough and swallowing outcomes, swallowing needs to be included in the primary aims. Please provide more details about the VFSS protocol e.g. number of boluses, consistencies, etc. This may help to explain why you did not find an association between PCF and PAS.
Authors’ response: All subjects underwent a formal instrumental swallowing assessment study. The rationale for performing the swallowing assessment were added in the introduction section as follows:
“…attempt to identify whether distinctive brain lesions are associated with poor VC in post-stroke patients; and second, to describe the swallowing level in those with low PCF during VC.”

The VFSS was performed according to the Logemann protocol. The details were provided:
“The FOIS were obtained at the time of the VFSS, which was performed according to standard protocols [35]. For the PAS, the worst PAS score across all boluses was used for analysis [36]. All assessments were performed within the first two weeks of acute stroke.”
Usage of the worst PAS score in any bolus may have been one of reasons for the poor correlation between the PCF and PAS, and this was also included in the limitation section.

Edited page: Page 4; line 181-183
Edited page: Page 11; line 369-370

Results
11.>Table 1 – please indicate whether the cough <80 L/min and the cough >80 L/min groups were significantly different on the variables presented in the table.
Authors’ response: We have indicated what variables showed significant differences by adding the P-values for Table 1. The PCF for the voluntary cough <80 L/min and the >80 L/min were corrected to “PCF <80 versus ≥ 80 L/min”
Edited page: Page 5

12. > P6 line 200: Among patients who developed AP within the first month, 52.6% had a PCF < 80L/min. Does that mean that the other 57.4% had PCF > 80 L/min? I’m not sure what the “versus 10%” is referring to.
Authors’ response: The authors have clarified this sentence as follows:
Those with PCF < 80 L/min (52.6%) showed a higher proportion of patients with aspiration pneumonia than those with ≥ 80 L/min (10%) within the first month after onset (p=0.004).
Edited page: Page 5; line 245-246

13. >Should PAS also be included in figure 3?
Authors’ response: We have included the PAS in both figure 3 and 4.
Edited page: Page 8-9

Discussion
14. > P8 line 239: I don’t agree with the authors’ conclusion that “results suggest that specific supratentorial lesions are associated with decreased VC function in patients diagnosed with stroke”, simply because only patients with subjectively weak VC were included in the analysis. It remains unknown whether, if patients with subjectively strong VC were also included, the same results would have been found. Please temper your conclusions accordingly.
Authors’ response: We agree with the reviewer’s comment. We tempered down our conclusions as follows: “The results of this VLSM study suggest that specific supratentorial lesions may be linked with decreased subjective cough weakness in patients diagnosed with stroke.”
Edited page: Page 9; line 286-288
15. > P8 line 249: “They also share the same afferents and efferents” – this is not correct, as one can produce a VC in the absence of sensory stimulation.
Authors’ response: This sentence was adapted from one of the references, but we acknowledge with the issues raised the reviewer. Therefore, we have made the corrections as follows: “They also share the same efferents.”

Edited page: Page 9; line 297-298

16.>P8 line 251: “The overlapping lesions include the temporal lobe and STG areas, which are also crucial components in the swallowing mechanism” – please provide a reference
Authors’ response: The reference was cited from Suntrup-Krueger et al , from “the impact of lesion location on dysphagia incidence, pattern and complications in acute stroke. Part 2: oropharyngeal residue, swallow and cough response, and pneumonia. Eur. J. Neurol. 2017, 24, 867-874.”
We have made the appropriate citation to the discussion section.

Edited Page: Page 10; line 300-301

17.>A major limitation of this work is that only patients who were subjectively identified as having weak VC were included in the study, thus creating a potential exclusion bias. Clinical judgement of cough strength is not strong (Laciuga et al., 2016; Miles & Hucakbee, 20130. Please note this as a limitation in this section.
Authors’ response: We agree with the limitations pointed by the reviewer and have thus made the following comments to the limitations section as follows:
“Subjective judgment of cough strength can be misleading [47,48] and prospective studies with large sample sizes that include those with no subjective cough weakness are warranted.”

Edited Page: Page 11; line 372-373

18.> Another limitation is the use of FOIS as a measure of dysphagia severity. At the time of swallowing assessment, the median FOIS score equated to nil by mouth status – this is more reflective of subjective clinical judgement than actual swallowing ability, particularly as these patients were measured prior to VFSS. Please consider removing the FOIS as an outcome or acknowledge that validity of the scale is grossly limited in acute stroke.
Authors’ response: The FOIS was measured at the time of the VFSS. We are in agreement with the limitations of the FOIS and have thus also included the GUSS and MASA. The short fallings of FOIS in acute stroke were commented in the discussion section.

“Although the FOIS [33] was assessed at the time as VFSS, caution is needed in judging patients’ swallowing abilities based solely on this parameter, especially in acute stroke patients. To overcome this factor, the GUSS [30,31] and MASA scales [32,33,37], which show good diagnostic properties in acute stroke, were included in the study.”
Edited Page: Page 11; line 397-399

We are grateful for the comments raised by the reviewer and hope that the revision addresses all the issues and concerns raised.

We are grateful for the comments raised by the reviewer and hope that the revision addresses all the issues and concerns raised.

Reviewer 2 Report

Thank you very much for this opportunity to review this manuscript. The aims and main contributions of this study were to investigate the effects of stroke lesions on voluntary cough (VC) function using voxel-based lesion-symptom mapping (VLSM), as well as to identify the brain lesions that are associated with poor VC in post-stroke patients.

The manuscript is accurately written and compels scientific interest. Nevertheless, this reviewer thinks it might benefit from English language editing, to make sure language accuracy.  

Please review the following sentence (lines 70-71):

“Only those with ischemic stroke confined to the supratentorial are and who met the following inclusion criteria were included for the analysis:”

Author Response

September 3rd, 2020

The following are the responses to the comments raised by the Reviewer 2
Dear reviewer;
We appreciate the comments raised during the review process. We have prepared a point-by-point response to the comments raised. The revised parts are marked with yellow highlight in the manuscript.

Thank you very much for this opportunity to review this manuscript. The aims and main contributions of this study were to investigate the effects of stroke lesions on voluntary cough (VC) function using voxel-based lesion-symptom mapping (VLSM), as well as to identify the brain lesions that are associated with poor VC in post-stroke patients.
1.>The manuscript is accurately written and compels scientific interest. Nevertheless, this reviewer thinks it might benefit from English language editing, to make sure language accuracy.
Authors’ response: We have received formal English editing service.
2.>Please review the following sentence (lines 70-71):
“Only those with ischemic stroke confined to the supratentorial are and who met the following inclusion criteria were included for the analysis:”
Authors’ response: Thank you for your invaluable comments. We have made the appropriate corrections as follows: “…only those with ischemic stroke confined to the supratentorial region who met the following inclusion criteria were recruited into the study.”
Edited Page 2; line 93-94

We are grateful for the comments raised by the reviewer and hope that the revision addresses all the issues and concerns raised.

Reviewer 3 Report

Thank you for the opportunity to review this manuscript. The aim of this retrospective, cross-sectional study was to examine the effects of brain lesions on voluntary cough in patients with supratentorial stroke. This study provides study provides robust and interesting findings that contribute to our understanding regarding the neurologic underpinnings associated with (impaired) voluntary cough response, and confirm previously established relationships between cough and swallowing. This is a strong manuscript, despite being retrospective in nature. The researchers are commended for their work, but are encouraged to address the below comments.

Abstract

  • Line 13: a subtle semantic change I recommend making is changing the phrase “…decrease cough force, which may increase the risk of aspiration” to “…decrease cough force, which is associated with an increased risk of aspiration”. This is because having impaired cough strength doesn’t in and of itself lead to aspiration. Instead, decreased cough strength is associated with aspiration risk (from a screening perspective) and is related to risks associated with aspiration (e.g., development of pneumonia).

Introduction

  • Line 57: it is a bit unclear how the first and second aims differ. Please make the difference clearer, or consider collapsing into a single aim.
  • The study includes additional comparisons between PCF and swallowing outcomes. Please include this as a secondary aim.

Methods

  • Line 65: what was the criteria for “weak cough”. What were there qualities used to determine that a cough sounded weak (if any)? Also, what were the cut-off values used in the objective analysis to confirm the presence of a weak cough?
  • Line 71: change “are” to “area”
  • Line 72: why was first-ever bilateral stroke not included?
  • Line 123: please indicate if this was a single voluntary cough or a sequential voluntary cough. Please also indicate if the patients coughed into an intraoral mouthpiece or a facemask and manufacture details of the spirometry equipment used.
  • Please describe the protocol used for the videofluoroscopic swallow study, and from what bolus/boluses the PAS was derived.
  • Was the FOIS derived from the videofluoroscopic swallow study, from the clinical assessment, or from patient report?
  • It is a bit unclear as to why swallowing outcomes were included given that this was not included in the introductory aims. Please include as an additional aim, or remove.
  • The use of an 80 L/minute cut-off value is peculiar for two reasons.
    • First, voluntary cough typically results in higher PCF than reflex cough. Therefore, applying reflex PCF cutoff values for aspiration pneumonia risk to voluntary cough seems potentially inappropriate – especially since cutoff values for voluntary cough have been previously explored. Would results have changed if using aspiration pneumonia cutoff values previously determined for voluntary cough (e.g., 400 L/minute – Kulnik et al., 2016; 242 L/minute – Bianchi et al., 2012). Please explain why voluntary cough cutoff values were not used?
    • Second, the article referenced for the 80 L/minute actually cites that 59 L/minute is the optimal cutoff. Please explain in greater detail why 80 L/minute was chosen as opposed to the 59 L/minute cutoff value that the authors are referencing. Would results have changed if 59 was used?

Results

  • Please provide intra- and inter-rater reliability of all outcome measures, as able

Discussion

  • There is no discussion of the relationship between PCF and swallowing/dysphagia in the Discussion section. This should be addressed, even if just a single paragraph, given the attention paid to swallowing in both the Methods and Results.
  • Line 313: stating that PAS is not a predictor of respiratory complication is a strong statement.  Many considerations should be included when suggesting such a finding, including the frequency of various PAS scores, the severity of various PAS scores, and on what bolus consistencies/volumes PAS scores were made. All of this should be included in the Methods, and incorporated into the Discussion if making such a statement.

Author Response

September 3rd, 2020
The following are the responses to the comments raised by the Reviewer 3
Dear reviewer
We appreciate the comments raised during the review process. We have prepared a point by point response to the comments raised. The revised parts are marked with underline in the manuscript.

1.>Line 13: a subtle semantic change I recommend making is changing the phrase “…decrease cough force, which may increase the risk of aspiration” to “…decrease cough force, which is associated with an increased risk of aspiration”. This is because having impaired cough strength doesn’t in and of itself lead to aspiration. Instead, decreased cough strength is associated with aspiration risk (from a screening perspective) and is related to risks associated with aspiration (e.g., development of pneumonia).
Authors’ response: We would to thank the reviewer for the comment. We agree with the need to change to “…decrease cough force, which is associated with an increased risk of aspiration”.
Edited page: Page 1; line14

2. >Line 57: it is a bit unclear how the first and second aims differ. Please make the difference clearer or consider collapsing into a single aim. The study includes additional comparisons between PCF and swallowing outcomes. Please include this as a secondary aim.
Authors’ response: We agree the need to present the hypothesis and aims of this study and we have made the following additions to the introduction:
“…we hypothesized that VC would be associated with specific supramedullary brain regions that have been linked with cortico-respiratory projections. In light of the close relationship between VC and swallowing, we also sought to determine whether low VC can lead to more pronounced difficulties in swallowing parameters. Thus, the aims of this study were to investigate the effects of stroke lesions on VC function using VLSM, determine whether distinctive brain lesions are associated with poor VC in post-stroke patients, and describe the swallowing levels in those with low PCF during VC.”
Edited page: Page 2; line 76-81

3. >Line 65: what was the criteria for “weak cough”. What were there qualities used to determine that a cough sounded weak (if any)? Also, what were the cut-off values used in the objective analysis to confirm the presence of a weak cough?
Authors’ response: Weak cough was defined as those “…who showed weak cough or decreased ability to clear the throat…” The objective cut-off values of a weak cough were defined as those with PCF less than 80 L/min, as assessed from a portable spirometer. The rationale of using this cut-off value would be discussed in more detail to the response for comment 9 and 10.
Edited page: Page 2; line 86
4. >Line 71: change “are” to “area”
Authors’ response: We have made the appropriate changes to “supratentorial region”.
Edited page: Page 2; line 94

5. >Line 72: why was first-ever bilateral stroke not included?
Authors’ response: As we have added in the methods, the main rationale were “…to meet the objectives of this study, only a homogenous group of ischemic stroke patients were included and those with intracerebral hemorrhage and bilateral brain lesions were excluded, as was carried out by previous VLSM studies. The edema and brain shifting in the former group and the more severe neurological deficits in the latter group can mislead VLSM analysis” Bilateral stroke lesions were excluded in reference to previous VLSM stroke studies.
Edited page: Page 3; line 104-107

6. > Line 123: please indicate if this was a single voluntary cough or a sequential voluntary cough. Please also indicate if the patients coughed into an intraoral mouthpiece or a facemask and manufacture details of the spirometry equipment used.
Authors’ response: For VC, the PCF value for each patient was presented as the mean of the three highest values over five consecutive attempts. The details of the equipment were included in the manuscript “ All subjects were asked to perform a quick, short, and explosive cough on a peak flow meter (Micropeak; Carefusion, Corp., San Diego, CA, USA) to a face mask” .
Edited page: Page 3 line 157-158
Edited page: Page 3 line 159-168

7.> Please describe the protocol used for the videofluoroscopic swallow study, and from what bolus/boluses the PAS was derived.Was the FOIS derived from the videofluoroscopic swallow study, from the clinical assessment, or from patient report?’

Authors’ response: The FOIS was derived from the VFSS. The VFSS were performed according to the Logemann protocol. The PAS was recorded as the worst PAS score across all tested boluses. Though there are still discussions on how to incorporate the PAS into the statistical analysis, the use of the worst PAS score was often used as an outcome parameter in previous studies. The appropriate references were included in the revised manuscript.
Edited page: Page 3 line 182-188

8.> It is a bit unclear as to why swallowing outcomes were included given that this was not included in the introductory aims. Please include as an additional aim, or remove.
Authors’ response: We agree with the reviewer’s comment and have included an additional aim at the introduction section, as explained earlier with additional objectives of “..we also sought to determine whether low VC can lead to more pronounced difficulties in swallowing parameters”and “…and describe the swallowing levels in those with low PCF during VC”
Edited page: Page 1; line 76-81

9.>The use of an 80 L/minute cut-off value is peculiar for two reasons. First, voluntary cough typically results in higher PCF than reflex cough. Therefore, applying reflex PCF cutoff values for aspiration pneumonia risk to voluntary cough seems potentially inappropriate – especially since cutoff values for voluntary cough have been previously explored. Would results have changed if using aspiration pneumonia cutoff values previously determined for voluntary cough (e.g., 400 L/minute – Kulnik et al., 2016; 242 L/minute – Bianchi et al., 2012). Please explain why voluntary cough cutoff values were not used? Second, the article referenced for the 80 L/minute actually cites that 59 L/minute is the optimal cutoff. Please explain in greater detail why 80 L/minute was chosen as opposed to the 59 L/minute cutoff value that the authors are referencing. Would results have changed if 59 was used?

Authors’ response: We would like to thank the reviewer for this important point. The voluntary cough is indeed higher than the reflexive cough. In the article that we had cited the 80L/min was the cut off value for the voluntary cough PCF, which showed good predictive values of respiratory complications. The 59 L/min was the PCF cut-off value from the reflexive cough obtained after citric acid inhalation.

We agree that this PCF VC was lower than those previously reported, which we commented on the limitation section, that “…. the lower values than those encountered in the previous report [46] may also be related to the higher age of patients in our group. However, because the PCF from VC are measured by volitional tests, with the patient making a maximal effort during the spirometry [51] the older age distribution might have limited this effort in our patients. Ageing has also been related to increased sarcopenia of the diaphragms [52], which in turn could lead to poor respiratory motor recruitment, causing insufficient coughs that would allow not full clearance.”
Edited page: Page 11; line 388-395

10.>Please provide intra- and inter-rater reliability of all outcome measures, as able
Authors’ response: We have provided for the intra and inter- rater measures of the swallowing parameters and were included in the manuscript. The diagnostic properties of these parameters have been demonstrated in previous studies; which have shown the excellent levels of inter-rater reliability of the MASA (Kappa = 0.82) [37], GUSS (agreement level = 83%)[30,31], and FOIS (agreement level = 85%)[34] scales, respectively.
Edited page: Page 4; line 186-188

11. > There is no discussion of the relationship between PCF and swallowing/dysphagia in the Discussion section. This should be addressed, even if just a single paragraph, given the attention paid to swallowing in both the Methods and Results.
Authors’ response: We agree with the reviewer comments and have added that
“… PCF values showed strong correlations with those with weak coughs showing more severe swallowing impairments as assessed by several parameters suggesting that swallowing and coughing are closely linked. One may cautiously suggest that the simultaneous involvement of some brain regions, such as the STG and SLF, could lead to both disordered swallowing poor production of a strong VC to clear the airways “,and that “ these results are similar to those of studies that have shown that low cough strength based on the measurement of cough flow can be used as an indicator of pneumonia risk and in acute stroke [46]. Respiration, swallowing, and coughing function all share similar neural and anatomical substrates.”
Edited page: Page 10; line 342-346, page 11; line 356-360

12. > Line 313: stating that PAS is not a predictor of respiratory complication is a strong statement. Many considerations should be included when suggesting such a finding, including the frequency of various PAS scores, the severity of various PAS scores, and on what bolus consistencies/volumes PAS scores were made. All of this should be included in the Methods, and incorporated into the Discussion if making such a statement.

Authors’ response: We agree with the issues raised by the reviewer concerning the PAS. In consideration that we incorporated the worst PAS during the VFSS, this might have contributed to the poor correlation between the PCF and PAS. Therefore, we have commented that “…second, the PCF and PAS scores showed a poor correlation. This may have been attributable to the fact that only patients with subjective weakness of cough were included in the analysis leading. In addition, since the worst PAS score was used for analysis, results should be interpreted with caution”
Edited page: Page 11; line 368-380

We are grateful for the comments raised by the reviewer and hope that the revision addresses all the issues and concerns raised.
